# Lemon Verbena Extract Enhances Sleep Quality and Duration via Modulation of Adenosine A1 and GABA_A_ Receptors in Pentobarbital-Induced and Polysomnography-Based Sleep Models

**DOI:** 10.3390/ijms26125723

**Published:** 2025-06-14

**Authors:** Mijoo Choi, Yean Kyoung Koo, Nayoung Kim, Yunjung Lee, Dong Joon Yim, SukJin Kim, Eunju Park, Soo-Jeung Park

**Affiliations:** 1Department of Food and Nutrition, Kyungnam University, Changwon 51767, Republic of Korea; mijoo@kyungnam.ac.kr (M.C.); kimnayoung99@student.kyungnam.ac.kr (N.K.); hjlee@kyungnam.ac.kr (Y.L.); 2Department of R&I Center, COMSMAXBIO, Seongnam 13486, Republic of Korea; ygkoo@cosmax.com (Y.K.K.); abmatics@naver.com (D.J.Y.); kimsukjin333@naver.com (S.K.); 3Department of Culinary Nutrition, Woo-Song University, Daejeon 34606, Republic of Korea

**Keywords:** lemon verbena, sleep quality, sleep duration, pentobarbital, electroencephalogram, electromyogram

## Abstract

This study investigated the effects of lemon verbena extract (LVE) on sleep regulation using both a pentobarbital-induced sleep model and an EEG-based sleep assessment model in mice. To elucidate its potential mechanisms, mice were randomly assigned to five groups: control, positive control (diazepam, 2 mg/kg b.w.), and three LVE-treated groups receiving 40, 80, or 160 mg/kg b.w. via oral administration. In the pentobarbital-induced sleep model, mice underwent a two-week oral administration of LVE, followed by intraperitoneal pentobarbital injections. The results demonstrated that LVE significantly shortened sleep latency and prolonged sleep duration compared to the control group. Notably, adenosine A1 receptor expression, both at the mRNA and protein levels, was markedly upregulated in the brains of LVE-treated mice. Furthermore, LVE’s administration led to a significant increase in the mRNA expression of gamma-aminobutyric acid type A (GABA_A_) receptor subunits (α2 and β2) in brain tissue. In the electroencephalography (EEG)/electromyogram (EMG)-based sleep model, mice underwent surgical implantation of EEG and EMG electrodes, followed by one week of LVE administration. Quantitative EEG analysis revealed that LVE treatment reduced wakefulness while significantly enhancing REM and NREM sleep’s duration, indicating its potential sleep-promoting effects. These findings suggest that LVE may serve as a promising natural sleep aid, improving both the quality and duration of sleep through the modulation of adenosine and GABAergic signaling pathways.

## 1. Introduction

Sleep is a fundamental physiological process crucial for maintaining overall health and well-being [1]. Disruptions in sleep patterns or insufficient sleep have been associated with cognitive impairment, mood disorders, and an elevated risk of chronic diseases such as cardiovascular disorders and metabolic syndromes [2]. As sleep disorders become increasingly prevalent, pharmaceutical interventions such as sedative-hypnotic drugs (e.g., benzodiazepines like diazepam and lorazepam, non-benzodiazepines such as zolpidem and eszopiclone, and melatonin receptor agonists like ramelteon) are commonly prescribed to alleviate symptoms [3]. However, prolonged use of these synthetic medications can lead to tolerance, dependence, and undesirable side effects, prompting a growing interest in natural alternatives for the enhancement of sleep, such as melatonin, valerian root, chamomile, and lemon balm. Among these, lemon verbena has gained attention for its potential sleep-promoting properties due to its rich content of natural antioxidants and calming phytochemicals [4].

Lemon verbena (*Aloysia citrodora*) is a medicinal plant traditionally used for its sedative and anxiolytic properties. This plant is known for its diverse bioactive properties, primarily attributed to its rich polyphenolic composition, particularly Verbascoside, which exhibits antioxidant, anti-inflammatory, and oxidative stress-modulating effects [5,6,7]. Recent studies have explored its bioactive compounds, particularly their potential effects on sleep regulation. Natural compounds that modulate neurotransmitter activity have gained attention for their role in improving sleep’s quality and duration [8,9]. Among the key neurochemical pathways involved in sleep regulation, the adenosine system and gamma-aminobutyric acid (GABA) signaling play crucial roles. The activation of the adenosine A1 receptor (A1R) has been shown to suppress neuronal excitability, thereby promoting sleep’s onset and prolonging sleep’s duration [10]. Additionally, GABA, the primary inhibitory neurotransmitter in the central nervous system, exerts sleep-promoting effects by binding to GABA_A_ receptor subunits, reducing neuronal excitability and inducing relaxation [11].

Previous studies have highlighted the efficacy of various natural compounds in modulating these pathways, yet comprehensive research on the sleep-promoting effects of *Aloysia citrodora* remains limited [12,13]. To bridge this gap, the present study investigates the effects of lemon verbena ethanol extract (LVE) on sleep regulation using both a pentobarbital-induced sleep model and an electroencephalography (EEG)-based sleep assessment model in mice. The pentobarbital-induced sleep model allows for the evaluation of the sedative effects of LVE by measuring sleep’s onset and duration under controlled conditions, providing insights into the compound’s potential to enhance sleep through the modulation of sleep–wake cycles. Among the various animal models used to evaluate the sleep-promoting effects of natural compounds, the pentobarbital-induced sleep model has been widely adopted as a standard pharmacological tool. Although pentobarbital is a barbiturate with anesthetic properties that acts by enhancing GABAergic transmission via the opening of chloride channels, it has long been utilized to assess sleep latency and total sleep duration due to its reliable and reproducible sedative effects. While the neural mechanisms of pentobarbital-induced sleep differ from those of physiological sleep, this model provides a convenient and consistent method to screen the sedative or sleep-enhancing properties of test substances in vivo [14,15]. In our study, we employed the pentobarbital-induced sleep model in mice to evaluate the potential sleep-promoting effects of lemon verbena extract (LVE), with a recognition of its limitations and in combination with EEG-based sleep architecture analysis. The EEG-based sleep model offers a more detailed understanding of the effects of LVE on sleep architecture, including the regulation of specific sleep stages such as NREM, REM, and wakefulness.

By evaluating LVE’s potential impact on sleep latency, sleep duration, and the expression of key sleep-related receptors, this study aims to explore possible mechanisms by which LVE may influence sleep quality. This study is intended to provide preliminary insights into the molecular pathways potentially involved in sleep regulation, with a particular focus on adenosine and GABAergic signaling, thereby contributing to the foundational understanding required for the future development of natural sleep aids. Additionally, the use of two distinct animal models strengthens the validity of the results and helps to paint a more comprehensive picture of LVE’s effects on sleep regulation. The findings from this study may contribute to broader research efforts investigating natural compounds for sleep enhancement and support further studies aimed at evaluating their applicability in the development of functional foods or supplements for managing sleep-related conditions.

## 2. Results

### 2.1. HPLC Analysis of Verbascoside in LVE

Verbascoside is one of the main compounds of LVE. The HPLC analysis of LVE revealed peaks matching those of the commercial standard Verbascoside, with retention times of 14~15 min (Figure 1). The LVE contained a minimum 24% of Verbascoside.

### 2.2. LVE Modulates Sleep Latency and Duration in Pentobarbital-Injected Mice

To assess the effect of oral LVE administration on sleep latency and duration, these parameters were measured in the pentobarbital-injected mice using the righting reflex test. Mice administered with DIZ and LVE exhibited a notable reduction in sleep latency compared to the NC group (*p* < 0.05; Figure 2A). The LVE 160 high-dose group showed a similar effect in reducing sleep latency as observed in the DIZ group. Additionally, LVE’s administration significantly prolonged the sleep duration in pentobarbital-injected mice compared to the NC group (*p* < 0.05; Figure 2B). The LVE 160 high-dose group also increased their sleep duration at a level similar to that observed in the DIZ group.

### 2.3. LVE Increases Serum Melatonin Levels and Brain GABA Concentrations in Pentobarbital-Injected Mice

Figure 3 shows the serum melatonin level and Brain GABA protein concentrations in the pentobarbital-injected mice. Oral administration of LVE at doses of 40, 80, and 160 mg/kg b.w. significantly increased serum melatonin levels in pentobarbital-injected mice compared to the NC group (*p* < 0.05; Figure 3A). This resulted in more effective outcomes compared to the positive control, the DIZ group. Notably, oral administration of LVE at doses of 40, 80, and 160 mg/kg b.w. significantly increased Brain GABA protein concentrations in pentobarbital-injected mice compared to the NC group (*p* < 0.05; Figure 3B). The LVE 160 group exhibited the highest GABA concentration, comparable to the effect observed in the DIZ group.

### 2.4. Effect of LVE on Adenosine A1 Receptor from Pentobarbital-Injected Mice

The mRNA expression of adenosine A1 receptor in the brains of pentobarbital-injected mice was assessed according to administered LVE doses of 40, 80, and 160 mg/kg b.w. Figure 4A shows that the expression of the adenosine A1 receptor in the DIZ group was significantly increased compared to the NC group. In contrast, the mRNA expression of adenosine A1 receptor in the LVE-treated groups (40, 80, and 160 mg/kg) was decreased compared to the DIZ group. However, no statistically significant differences were observed between the LVE 80 and LVE 160 mg/kg b.w. groups and the DIZ group.

### 2.5. LVE Modulates GABA_A_ Receptor Expression in the Brains of Pentobarbital-Injected Mice

To assess the effect of LVE intake on GABA_A_ receptor expression, the mRNA expression of the subunits (α1, α2, β1, β2, γ1, and γ2) in the brains of the pentobarbital-injected mice was measured. The mRNA expression of GABA_A_ receptor α1 in the LVE-administered groups (40, 80, and 160 mg/kg b.w.) was comparatively higher than that in the DIZ group; however, no statistically significant differences were observed among all groups (Figure 4B). The mRNA expression of GABA_A_ receptor α2 in the LVE-administered groups (40, 80, and 160 mg/kg b.w.) was significantly decreased compared to the DIZ group (Figure 4E). Moreover, no statistically significant differences were observed in the mRNA expressions levels of GABA_A_ receptor β1 and γ1 among all groups (Figure 4C,D). The mRNA expression of GABA_A_ receptor β2 in the DIZ group was significantly increased compared to the NC group. However, LVE treatment inhibited the mRNA expression of GABA_A_ receptor β2 compared to the DIZ group (Figure 4F). The mRNA expression of GABA_A_ receptor γ2 in the DIZ group was significantly decreased compared to the NC group. In the LVE-treated groups, the mRNA expression of GABA_A_ receptor γ2 was increased compared to the DIZ group (Figure 4G).

### 2.6. LVE Modulates GABA_A_ Receptor Protein Expression in the Brains of Pentobarbital-Injected Mice

To evaluate the effect of LVE intake on GABA_A_ receptor expression at the protein level, the protein expression of its subunits (α1, α2, β1, β2, γ1, and γ2) in the brains of the pentobarbital-injected mice was measured (Figure 5A). No statistically significant differences were observed in the protein expression levels of GABA_A_ receptor α1, β1, and γ1 across all groups (Figure 5B–D). The protein expression of GABA_A_ receptor α2 and β2 in the DIZ group was significantly increased compared to the NC group. In contrast, the protein expression of GABA_A_ receptor α2 and β2 in the LVE-treated groups showed a related decrease compared to the DIZ group (Figure 5E,F). In the case of the protein expression of GABA_A_ receptor γ2, the results were opposite to those observed for GABA_A_ receptor α2 and β2. The protein expression of GABA_A_ receptor γ2 in the LVE-treated groups was significantly increased compared to the DIZ group (Figure 5G).

### 2.7. Body Weight Changes in C57BL/6N Mice Related to EEG/EMG-Based Sleep Models

Pre-surgery weight comparisons among all groups revealed no statistically significant differences (Table 1).

### 2.8. LVE Improves Sleep Quality and Duration in EEG/EMG-Based Sleep Models

Changes in sleep duration following each sample intake were analyzed through quantitative EEG/EMG analysis (Figure 6). The wake time in the DIZ, KA001, KB002, and KC003 groups was significantly reduced compared to baseline (*p* < 0.001), with KA001 showing a significant increase in both REM and NREM sleep durations compared to baseline (*p* < 0.001). The NREM sleep duration in the DIZ, KB002, and KC003 groups was significantly increased compared to baseline, although no significant differences were observed in REM sleep duration (Figure 7A). When the baseline EEG quantitative analysis values were set as 100%, the wake time in the DIZ, KA001, KB002, and KC003 groups decreased by 60.2%, 52.8%, 55.9%, and 55.4%, respectively (*p* < 0.001). Notably, KA001 and KB002 showed significant increases in REM sleep time by 73.6% and 44.3%, respectively, and in NREM sleep time by 110.7% and 194.5%, respectively. While DIZ and KC003 exhibited significant increases in NREM sleep time by 147.1% and 154.7%, respectively (*p* < 0.001), no significant differences were observed in REM sleep duration (Figure 7B).

Comparing the differences by sample, the wake time in the DIZ, KA001, KB002, and KC003 groups decreased compared to baseline, while REM sleep duration was increased in KA001 and KB002. The NREM sleep time in DIZ, KA001, KB002, and KC003 increased relative to baseline (Figure 8). EMG integral values were significantly decreased by 25.3%, 34.6%, 18.0%, and 18.7% in the DIZ, KA001, KB002, and KC003 groups, respectively, compared to baseline (*p* < 0.001) (Figure 9). To further evaluate sleep quality following sample administration, FFT delta power analysis was performed (Figure 10). Compared to baseline, all treatment groups exhibited a significant increase in delta power, with values reaching 140.0% in the DIZ group, 149.0% in the KA001 group, 144.0% in the KB002 group, and 149.0% in the KC003 group (*p* < 0.001).

## 3. Discussion

The regulation of sleep involves intricate interactions between circadian rhythms, sleep homeostasis, and external influences [16]. Sleep homeostasis is responsible for balancing the sleep–wake cycle by increasing the drive for sleep following extended wakefulness and promoting wakefulness after prolonged sleep [17]. The suprachiasmatic nucleus of the hypothalamus serves as the main circadian pacemaker, controlling the synchronization of sleep–wake patterns in response to environmental factors [18]. This study aimed to explore the potential of LVE and its active compound, Verbascoside, as a herbal alternative for enhancing sleep. Previous research on LVE has indicated that it may possess sleep-promoting properties [19,20,21], but the underlying mechanisms have yet to be thoroughly investigated.

*Aloysia citrodora* (lemon verbena), a member of the Verbenaceae family, is a flowering plant native to South America that is now widely cultivated in various regions, including the Middle East and the Mediterranean [22,23,24]. While research on its sedative potential remains limited, some clinical studies have explored its ability to alleviate sleep disturbances and reduce anxiety-related symptoms [25,26]. Experimental findings suggest that *A. citrodora* may exert its sleep-enhancing effects through the modulation of the GABAergic system, potentially mimicking benzodiazepine-like activity by interacting with GABA receptors [27,28].

In a previous clinical trial [29], sleep quality was assessed using various evaluation tools, including the Pittsburgh Sleep Quality Index (PSQI). The results showed that LVE intake led to improvements in overall sleep efficiency, including an increased sleep duration and reduced sleep onset latency. However, the mechanisms underlying these effects remained unclear. In the present study, we focused on elucidating the mechanisms by which LVE may modulate sleep regulation. Specifically, we examined the role of adenosine A1 receptors and GABA_A_ receptors in the sleep-enhancing effects of LVE. Our results revealed that oral administration of *LVE* significantly increased the mRNA expression of adenosine A1 receptors in pentobarbital-induced sleep in mice. Adenosine receptors, particularly A1 and A2 subtypes, are essential for regulating sleep and wakefulness by modulating excitatory and inhibitory neurotransmission [30]. The activation of A1 receptors has been linked to antidepressant-like effects and an improved sleep quality [31,32], suggesting that *LVE* may contribute to sleep regulation through adenosine A1 receptor activation.

In addition to adenosine receptors, GABAergic signaling plays a central role in sleep regulation. GABA_A_ receptors, in particular, are implicated in sleep’s initiation and maintenance [33]. GABA_A_ receptor agonists, such as sedatives and anesthetics, act as positive allosteric modulators, enhancing GABA’s inhibitory effects to induce sleep [34]. The GABA_A_ receptor is a pentameric complex composed of various subunits, with the most studied configuration consisting of α, β, and γ subunits [35]. In our study, we observed a significant increase in the mRNA expression of GABA_A_ receptor subunits α2 and β2 in the brains of mice administered *LVE*. However, there were no significant changes in the mRNA expression of GABA_A_ receptor subunits α1, β1, and γ1, indicating that *LVE* may selectively influence specific subunits of the GABA_A_ receptor.

In our study, the mRNA and protein expression levels of the GABA_A_ receptor γ2 subunit were significantly reduced in both the DIZ and LVE-treated groups compared to the control group. The γ2 subunit, which contains the benzodiazepine-binding site, is crucial for GABAergic neurotransmission [36]. The reduction in γ2 subunit expression suggests that LVE may promote sleep through a mechanism distinct from benzodiazepines, possibly by selectively modulating other GABA_A_ receptor subtypes, as indicated by the upregulation of the α2 and β2 subunits. This suggests that LVE may facilitate sleep by naturally modulating GABAergic signaling, rather than enhancing benzodiazepine-sensitive receptor activity. Additionally, the γ2 subunit is involved in synaptic stabilization and clustering, and its downregulation may reflect an adaptive response to prolonged receptor activation, shifting the balance toward a sustained inhibitory tone, rather than excessive GABAergic potentiation [37].

Melatonin is widely recognized as a modulator involved in the regulation of circadian rhythms and sleep; however, its precise role as a sleep regulator remains complex and subject to ongoing investigation [38,39]. In this study, the role of melatonin secretion in the sleep-promoting effects of LVE was found to be significant. Previous studies have demonstrated mixed results regarding the effect of melatonin on sleep, with some showing beneficial effects and others reporting only marginal improvements [40,41]. Our analysis revealed a significant difference in serum melatonin levels between *LVE*-treated and control mice, suggesting that the sleep-enhancing effects of lemon verbena may involve a complex interaction with melatonin secretion. Although serum melatonin levels alone may not directly confirm sleep enhancement, our study further demonstrated that LVE significantly upregulated adenosine A1 receptor expression and GABA_A_ receptor subunits (α2 and β2) in the brain. Additionally, in the EEG-based sleep model, LVE treatment significantly increased REM and NREM sleep duration while reducing wakefulness.

Furthermore, this study suggests that Verbascoside, a key bioactive compound in LVE, may act as a potential modulator of sleep. Based on the observed results, it is inferred that Verbascoside could contribute to the sleep-promoting effects of LVE, potentially modulating the adenosine and GABAergic systems to enhance sleep duration and reduce sleep latency. Verbascoside has been previously studied for its antioxidant, anti-inflammatory, and neuroprotective effects [42,43,44]. Our findings suggest that Verbascoside may contribute to the sleep-promoting effects of *LVE* by modulating both the adenosine and GABAergic systems. The active compound may work synergistically with other components of lemon verbena to enhance sleep duration and reduce sleep latency.

In this study, we employed an EEG/EMG-based sleep model to evaluate the sleep-modulating effects of LVE. The EEG/EMG model allowed for a precise analysis of sleep architecture, including sleep’s onset, sleep stages (REM, NREM), and wakefulness, offering valuable insights into how LVE influences the sleep–wake cycle. The results from our EEG/EMG analysis revealed that LVE’s administration significantly increased the total sleep duration, particularly enhancing NREM sleep and REM sleep while reducing wakefulness. The sleep onset latency was also notably shorter in the LVE-treated group compared to the control, suggesting a potential facilitatory effect of LVE on sleep’s initiation. These changes were consistent across both REM and NREM stages, indicating that LVE may promote a more consolidated and restorative sleep profile. These findings corroborate the improvements in sleep quality previously demonstrated in human clinical trials [29].

It should be noted that while pentobarbital-induced sleep models are widely used to assess the sedative and hypnotic properties of compounds due to their simplicity and reproducibility, this pharmacological model primarily reflects anesthetic rather than physiological sleep states. The hypnotic effects induced by pentobarbital differ fundamentally from natural sleep mechanisms and involve different neural pathways regulating sleep architecture [45,46,47]. To address this limitation and provide more robust evidence, our study complemented behavioral observations with EEG/EMG analysis. This dual approach allowed us to verify sleep’s latency, duration, and structure, including NREM and REM stages, thereby enhancing the physiological relevance and reliability of our findings regarding LVE’s sleep-promoting effects. By integrating behavioral and electrophysiological assessments, we aimed to mitigate concerns about the interpretation of pentobarbital model data and offer a more comprehensive evaluation of LVE’s therapeutic potential.

Interestingly, unlike pharmaceutical sedatives that primarily act through the direct modulation of the GABAergic system [48], our data suggest that LVE’s effects may stem from a more natural modulation of sleep-related pathways. This is supported by the significant increases observed in NREM and REM sleep and the reduction in wakefulness, which align with a balanced enhancement of both restorative sleep and sleep efficiency. Additionally, the reduction in wakefulness suggests that LVE may play a role in enhancing sleep stability and prolonging restful sleep periods, potentially through modulation of neurotransmitter systems involved in arousal and sleep regulation. These results support the idea that LVE could offer a safer, non-addictive alternative to conventional sleep aids, which often have side effects associated with long-term use.

While the current study provides compelling evidence for the sleep quality-enhancing effects of LVE, several aspects present valuable opportunities for future exploration. First, although we observed notable changes in serum melatonin levels, we did not directly assess melatonin receptor activity in the central nervous system. Incorporating receptor binding assays in future studies could offer deeper insights into the potential involvement of melatonergic signaling in LVE’s effects and further validate its mechanism of action. Second, while the pentobarbital-induced sleep model provided clear evidence of LVE’s sedative properties, additional behavioral assays—such as anxiety-related tests or alternative sleep initiation models—could broaden our understanding of LVE’s neurobehavioral profile, including its potential anxiolytic effects. Third, although the current study highlights the beneficial effects of LVE, we did not compare its efficacy with that of other natural compounds known for sleep-promoting properties. Including such comparisons in future studies could offer valuable insights into the relative therapeutic potential of LVE and help position it more accurately within the landscape of natural sleep aids. Addressing these limitations in future research will further elucidate the therapeutic potential of lemon verbena extract as a natural sleep aid.

## 4. Materials and Methods

### 4.1. Preparation of Lemon Verbena Extract and High-Performance Liquid Chromatography (HPLC)

The LVE used in the main experiment was manufactured by Monteloeder SL (Alicante, Spain) and supplied by COMSMAXBIO (Seongnam, Republic of Korea). The LVE underwent purification, filtration, concentration, clarification, and vacuum-drying to yield a refined LVE product. HPLC analysis was conducted using a 1260 Infinity II LC System (Agilent Technologies Spain, S.L., Las Rozas, Madrid, Spain) equipped with a BDS Hypersil C18 column (250 × 4.6 mm, 5 µm particle size; Thermo Scientific, Waltham, MA, USA). The detection wavelength was set to 330 nm. The mobile phase consisted of acetic acid and acetonitrile, with a flow rate of 1 mL/min. A sample volume of 20 μL was injected for each analysis. Verbascoside was identified as the marker compound, with a retention time (RT) of approximately 14–15 min.

### 4.2. Pentobarbital-Induced Sleep in Mice

ICR mice (female, 4 weeks old) were purchased from Japan SLC, Inc. (Hamamatsu, Shizuoka, Japan) and maintained in a controlled environment under a 12 h light/dark cycle, with a temperature of 22 ± 2 °C and humidity levels ranging from 50% to 60%. This strain and sex are known to show a higher sensitivity to pharmacological stimuli, which is advantageous for measuring drug-induced sleep’s onset and duration. ICR mice are genetically diverse and widely used in drug screening studies due to their robust and consistent responses. Moreover, female mice are generally less aggressive and exhibit reduced stress-related variability, making them more suitable for acute behavioral assays such as sleep duration measurements [49]. After a 7-day acclimation period on an AIN-93G Growth Purified Diet (Catalog #57W5, TestDiet^®^, Richmond, IN, USA), the mice were randomly divided into five groups (*n* = 8): a normal control group (NC), a positive control group receiving diazepam (DIZ, 2 mg/kg b.w., i.p.), and three experimental groups administered LVE orally at doses of 40, 80, and 160 mg/kg b.w. LVE was administered orally to reflect the typical consumption of natural sleep-enhancing supplements. To induce sleep, pentobarbital (a commonly used sedative for sleep’s induction) was injected intraperitoneally (i.p.) 45 mg/kg b.w. post-administration [29,50]. After two weeks of oral treatment and intraperitoneal injections, the mice were euthanized by cervical dislocation. Blood was collected from the abdominal aorta and allowed to clot at room temperature for 30 min, followed by centrifugation at 3000× *g* for 15 min to separate serum. Brain tissues were immediately collected for used in the RT-PCR and Western blot assay, as described in their respective method sections, following the guidelines approved by the Animal Care and Use Review Committee of Kyungnam University (KUIAC_24_05).

### 4.3. Sleep Latency and Sleep Duration

Sleep latency was defined as the interval between pentobarbital’s administration and the loss of the righting reflex. Total sleep duration was determined by measuring the time from the loss of the righting reflex to its subsequent reappearance.

### 4.4. ELISA

Serum melatonin levels in pentobarbital-treated mice were quantified using a melatonin ELISA kit (BioVisioin, Inc., Milpitas, CA, USA). Brain GABA concentrations were measured using the Mouse Gamma-Aminobutyric Acid kit (MyBiosource, San Diego, CA, USA). All biochemical markers were analyzed according to the manufacturer’s protocol.

### 4.5. Real-Time Polymerase Chain Reaction (RT-PCR)

mRNA was extracted from brain tissues, including both the cortex and thalamus, using the RNeasy Mini Kit (QIAGEN, Germantown, MD, USA). cDNA synthesis was then carried out using the iScript^TM^ cDNA Synthesis Kit (Bio-Rad Laboratories, Hercules, CA, USA). Subsquently, RT-PCR was performed with SYBR Green PCR Master Mix (IQ SYBR Green Supermix; Bio-Rad Laboratories). The amplification conditions involved 40 cycles of denaturation at 95 °C for 15 s, annealing at 60 °C for 30 s, and extension at 72 °C for 45 s, using primers referenced in previous studies [51]. The mRNA expression levels of adenosine A1 receptor, GABA_A_ receptor α1, GABA_A_ receptor α2, GABA_A_ receptor β1, GABA_A_ receptor β2, GABA_A_ receptor γ1, GABA_A_ receptor γ2, and GAPDH were quantified. Data analysis was conducted with the 7500 System SDS software version 1.3.1 (Applied Biosystems, Foster City, CA, USA).

### 4.6. Protein Extraction and Western Blot Analysis

For protein expression analysis in brain tissues, including the cortex and thalamus, equal amounts of total protein were separated using sodium dodecyl sulfate polyacrylamide gel electrophoresis (SDS PAGE) and then transferred to a nitrocellulose membrane. After transfer, the membranes were blocked with a blocking buffer at room temperature for 5 min. The membranes were then incubated with primary antibodies GABA_A_ receptor α1, GABA_A_ receptor α2, GABA_A_ receptor β1, GABA_A_ receptor β2, GABA_A_ receptor γ1, GABA_A_ receptor γ2, or β-actin (dilution 1:1000; Cell Signaling, Boson, MA, USA) overnight at 4 °C. Following primary antibody incubation, the membranes were treated with a secondary antibody (anti-rabbit IgG HRP-linked antibody, dilution 1:1000; Cell Signaling) for 1 h at room temperature. Protein bands were visualized using enhanced chemiluminescence (Bio-Rad Laboratories) and captured with the ChmiDoc XRS+ system (Bio-Rad Laboratories).

### 4.7. Electroencephalography (EEG) and Electromyogram (EMG) Implantation Surgery Model

C57BL/6N mice (male, 9 weeks old, total 32 mice) were purchased from Saeronbio, Inc. (Uiwang, Republic of Korea). For the EEG/EMG implantation and sleep stage analysis, we used C57BL/6N mice, a commonly used inbred strain in neuroscience research with a well-established sleep architecture. Male mice were selected to minimize variability due to hormonal cycles that may influence EEG readings. Additionally, nine-week-old mice are considered to be mature enough to undergo the surgical procedures required for electrode implantation and to recover reliably post-surgery [52,53]. The mice were housed in the same controlled environment under a 12 h light/dark cycle, with a temperature of 22 ± 2 °C and humidity levels ranging from 50% to 60%. After a 7-day acclimation period on an AIN 93G diet, the mice were anesthetized with an avertin solution and securely positioned in a stereotaxic frame (Model 963, Kopf Instruments, Tujunga, CA, USA). To maintain respiratory anesthesia, isoflurane was continuously administered throughout the procedure. Following a midline incision of the scalp, three electrode implantation sites were carefully selected and perforated using a micro-drill. Three EEG electrodes were implanted and firmly secured, while two EMG electrodes were inserted into the rhomboid muscle. After electrode placement, the incised area was sutured, and the surgical site was sealed with dental cement (3M RelyX Luting, 3M Co., Saint Paul, MN, USA). The mice were then removed from the stereotaxic frame and placed in a recovery cage for a five-day postoperative recovery period. Subsequently, the mice were acclimated for four days under conditions identical to those of the EEG recording environment.

In this EEG/EMG-based polysomnography study, all the animals were monitored for 24 h to collect baseline sleep data prior to the compound’s administration. This self-controlled, within-subject design is a well-established and reliable method in sleep research, as it reduces inter-individual variability and enables a more accurate evaluation of drug-induced alterations in sleep architecture. Furthermore, a diazepam-treated group was included as a positive control to validate the sensitivity of the model and provide a comparative reference for the effects of LVE. As each mouse served as its own control based on its baseline sleep profile, the absence of a separate vehicle-only group does not compromise the reliability or interpretability of the data. All experimental procedures were conducted in accordance with the guidelines approved by the Animal Care and Use Review Committee of Kyungnam University.

### 4.8. EEG Recording and Sleep–Wake State Analysis

Following the adaptation period in the EEG recording environment, EEG and EMG signals were recorded for 24 h starting at 13:00 using the EEG Logger-2 system (Bio Research Center Co., Ltd., Nagoya, Aichi, Japan). Baseline EEG signals were obtained by recording for 24 h prior to the sample’s administration. After baseline measurement, the mice were randomly assigned to experimental groups (4 groups, *n* = 8) and administered either diazepam 2 mg/kg/day (DIZ), LVE 40 mg/kg/day (KA001), LVE 80 mg/kg/day (KB002), or LVE 160 mg/kg/day (KC003) orally for 7 consecutive days. No animals, experimental units, or data points were excluded during the study. Subsequently, EEG and EMG recordings were performed for an additional 24 h. All recorded EEG data were analyzed using SleepSign software (ver. 3.1.0, Kissei Comtec, Nagano, Japan) to assess sleep–wake profiles [54]. The recordings were segmented into 10 s epochs and categorized into wakefulness, rapid eye movement (REM) sleep, and non-rapid eye movement (NREM) sleep. Sleep duration and architecture were specifically analyzed within an 8 h period corresponding to the dark (active) phase of the light–dark cycle, reflecting the time of peak sleep–wake cycling in mice. The results were extracted in 1 h intervals to evaluate sleep architecture during this biologically relevant period. The initial EEG and EMG data were analyzed using the automated scoring algorithm built into the SleepSign software. To ensure accuracy, particularly for REM sleep classification, a blinded researcher manually reviewed the entire 24 h recording and made corrections as needed. This manual validation step was implemented to minimize potential misclassification errors by the automated system.

### 4.9. Statistical Analysis

All data are expressed as the mean ± standard deviation (SD). Statistical analyses were performed using a one-way analysis of variance (ANOVA), followed by Duncan’s multiple range test for post hoc comparisons to determine significant differences among groups. And paired *t*-tests were used to compare the baseline and experimental group, in addition to the mean duration of runs for sleep and each sleep stage. Data analysis was conducted using SPSS PASW Statistics 22.0 (SPSS, Inc., Chicago, IL, USA), with a significance threshold set at *p* < 0.05.

## 5. Conclusions

In conclusion, this study provides compelling evidence for the sleep-promoting effects of lemon verbena extract (LVE) through the use of both pentobarbital-induced sleep in mice and an EEG/EMG-based polysomnography sleep model. The results from both animal models demonstrate that LVE significantly improves sleep duration, enhances sleep stages such as NREM and REM, and reduces wakefulness. These findings suggest that LVE may influence the sleep–wake cycle through the modulation of key neurotransmitter systems, particularly the adenosine A1 and GABA_A_ receptors. While a direct effect of Verbascoside, the active compound in LVE, on sleep’s promotion was not definitively confirmed in this study, its presence in the extract suggests it may contribute to the observed sleep-enhancing effects. We have previously confirmed the sleep quality improvement effects of lemon verbena extract through clinical trials, demonstrating its ability to shorten sleep latency and improve PSQI and actigraphy scores. In this study, we further demonstrated that lemon verbena extract is involved in the GABA signaling pathway by regulating GABA and adenosine A1 receptor genes. These findings suggest that the sleep quality improvement observed in clinical trials may be attributed to the GABA mechanism.

Given the promising effects of LVE on sleep, this study highlights its potential as a natural, safe alternative for improving sleep’s quality and duration. LVE may serve as a viable option for functional foods, dietary supplements, or even therapeutic agents for sleep disorders. However, further research is needed to investigate the precise molecular mechanisms through which Verbascoside and other bioactive compounds in LVE modulate sleep regulation. Identifying the specific neurochemical pathways involved could provide deeper insights into how LVE promotes sleep.

## Figures and Tables

**Figure 1 ijms-26-05723-f001:**
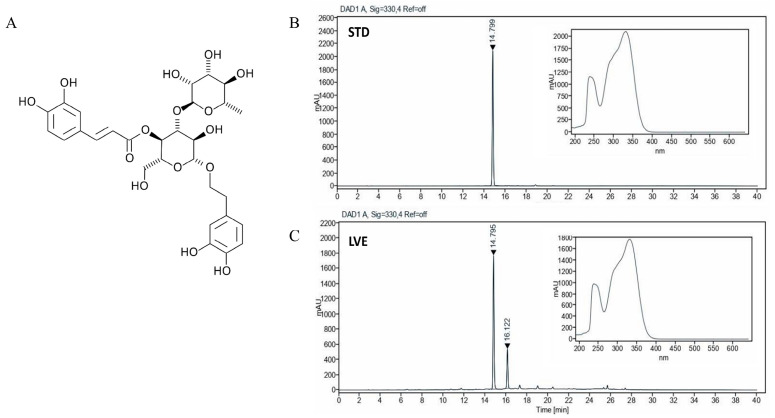
High-performance liquid chromatogram of Verbascoside in lemon verbena ethanol extract (LVE) at 330 nm. (**A**) Verbascoside structure, (**B**) Verbascoside standard chromatogram, (**C**) LVE chromatogram for verbascoside. Verbascoside appeared with retention times of approximately 14~15 min.

**Figure 2 ijms-26-05723-f002:**
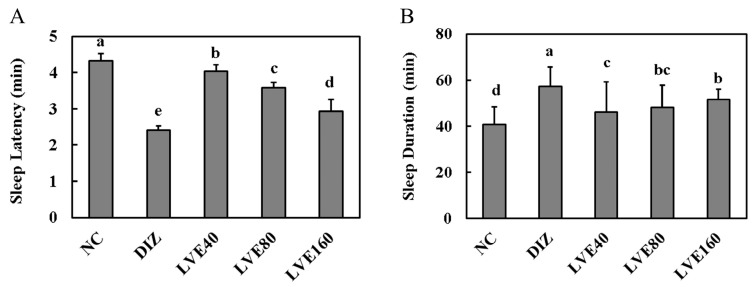
Lemon verbena ethanol extract (LVE) modulates sleep latency and duration in pentobarbital-injected mice. (**A**) Sleep latency (min); (**B**) sleep duration (min). NC, normal control; DIZ, Diazepam 2 mg/kg b.w.; LVE 40, LVE 40 mg/kg b.w.; LVE 80, LVE 80 mg/kg b.w.; LVE 160, LVE 160 mg/kg b.w. Oral administration for each group was continued for 2 weeks. Data are presented as mean ± standard deviation (SD). Statistical significance was determined using one-way ANOVA followed by post hoc test. Different letters (a, b, c, d, e) above bars indicate significant differences between groups (*p* < 0.05).

**Figure 3 ijms-26-05723-f003:**
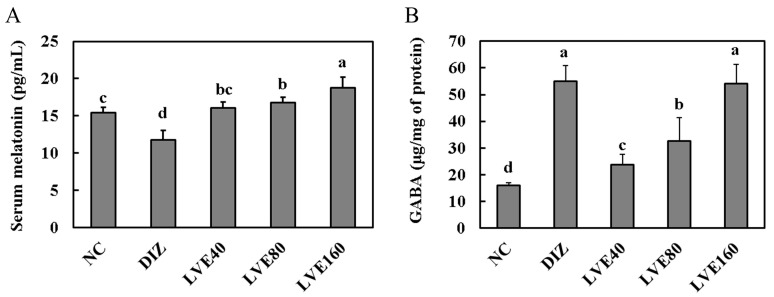
Lemon verbena ethanol extract (LVE) improves serum melatonin levels and Brain GABA concentrations in pentobarbital-injected mice. (**A**) Serum melatonin; (**B**) GABA concentration. NC, normal control; DIZ, Diazepam 2 mg/kg b.w.; LVE 40, LVE 40 mg/kg b.w.; LVE 80, LVE 80 mg/kg b.w.; LVE 160, LVE 160 mg/kg b.w. Oral administration for each group was continued for 2 weeks. Data are presented as mean ± standard deviation (SD). Statistical significance was determined using one-way ANOVA followed by post hoc test. Different letters (a, b, c, d) above bars indicate significant differences between groups (*p* < 0.05).

**Figure 4 ijms-26-05723-f004:**
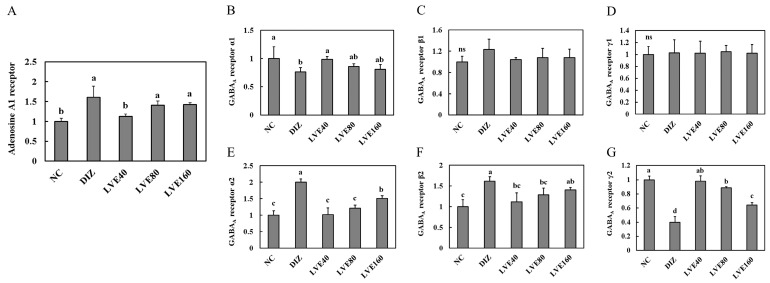
Lemon verbena ethanol extract (LVE) modulates adenosine A1 receptor and GABA_A_ receptor expression in the brains of pentobarbital-injected mice. (**A**) The mRNA expression of Adenosine A1 receptor; (**B**) the mRNA expression of GABA_A_ receptor α1; (**C**) the mRNA expression of GABA_A_ receptor β1; (**D**) the mRNA expression of GABA_A_ receptor γ1; (**E**) the mRNA expression of GABA_A_ receptor α2; (**F**) the mRNA expression of GABA_A_ receptor β2; (**G**) the mRNA expression of GABA_A_ receptor γ2. NC, normal control; DIZ, Diazepam 2 mg/kg b.w.; LVE 40, LVE 40 mg/kg b.w.; LVE 80, LVE 80 mg/kg b.w.; LVE 160, LVE 160 mg/kg b.w. Oral administration for each group was continued for 2 weeks. Data are presented as mean ± standard deviation (SD). Statistical significance was determined using one-way ANOVA followed by a post hoc test. Different letters (a, b, c, d) above the bars indicate significant differences between groups (*p* < 0.05). ns: not signficant.

**Figure 5 ijms-26-05723-f005:**
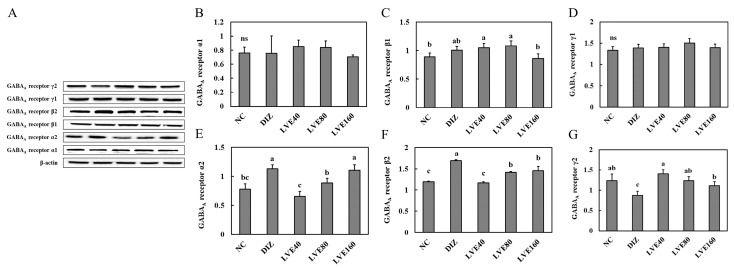
Lemon verbena ethanol extract (LVE) modulates GABA_A_ receptor protein expression in the brains of pentobarbital-injected mice. (**A**) Western blot analysis of GABA_A_ receptors-related protein expression. (**B**) Quantification of GABA_A_ receptor α1. (**C**) Quantification of GABA_A_ receptor β1. (**D**) Quantification of GABA_A_ receptor γ1. (**E**) Quantification of GABA_A_ receptor α2. (**F**) Quantification of GABA_A_ receptor β2. (**G**) Quantification of GABA_A_ receptor γ2. NC, normal control; DIZ, Diazepam 2 mg/kg b.w.; LVE 40, LVE 40 mg/kg b.w.; LVE 80, LVE 80 mg/kg b.w.; LVE 160, LVE 160 mg/kg b.w. Oral administration for each group was continued for 2 weeks. Data are presented as mean ± standard deviation (SD). Statistical significance was determined using one-way ANOVA followed by a post hoc test. Different letters (a, b, c) above the bars indicate significant differences between groups (*p* < 0.05). ns: not signficant.

**Figure 6 ijms-26-05723-f006:**
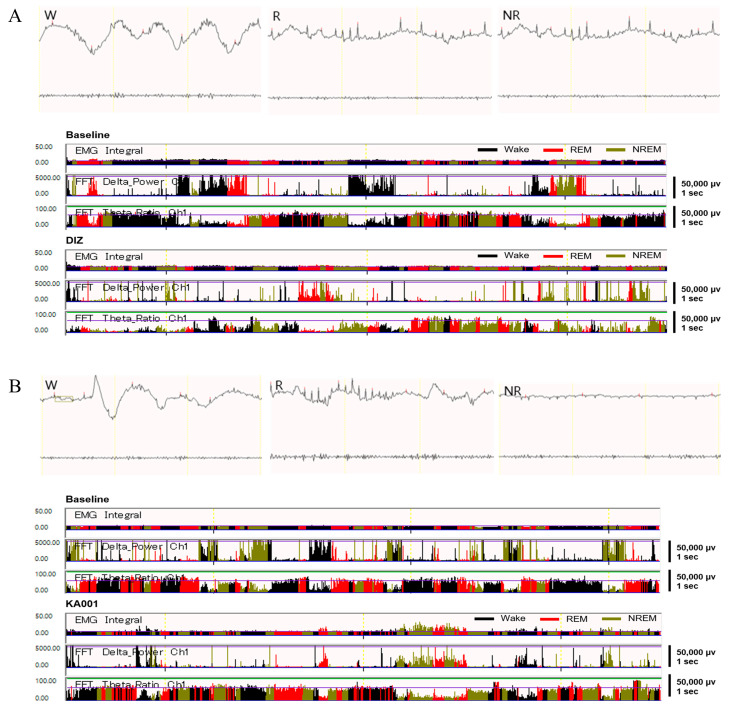
EEG and EMG recordings in polysomnography-based sleep models. Representative EEG and EMG traces and corresponding hypnograms are shown for each group: (**A**) Diazepam 2 mg/kg/day (DIZ), (**B**) LVE 40 mg/kg/day (KA001), (**C**) LVE 80 mg/kg/day (KB002), (**D**) LVE 160 mg/kg/day (KC003). Oral administration for each group was continued for 7 consecutive days, followed by a 24 h EEG/EMG recording session. In each panel, the top trace shows EEG waveforms across wake (W), rapid eye movement (REM, R), and non-rapid eye movement (NREM, NR) sleep stages. EMG integral, EEG delta power (0.5–4 Hz), and theta ratio (θ/δ) are plotted over time. Hypnograms below indicate the vigilance states (wake: black, REM sleep: red, NREM sleep: olive green). EEG signals were processed using fast Fourier transform (FFT) analysis to quantify spectral power. Scale bars: 50,000 μV, 1 s.

**Figure 7 ijms-26-05723-f007:**
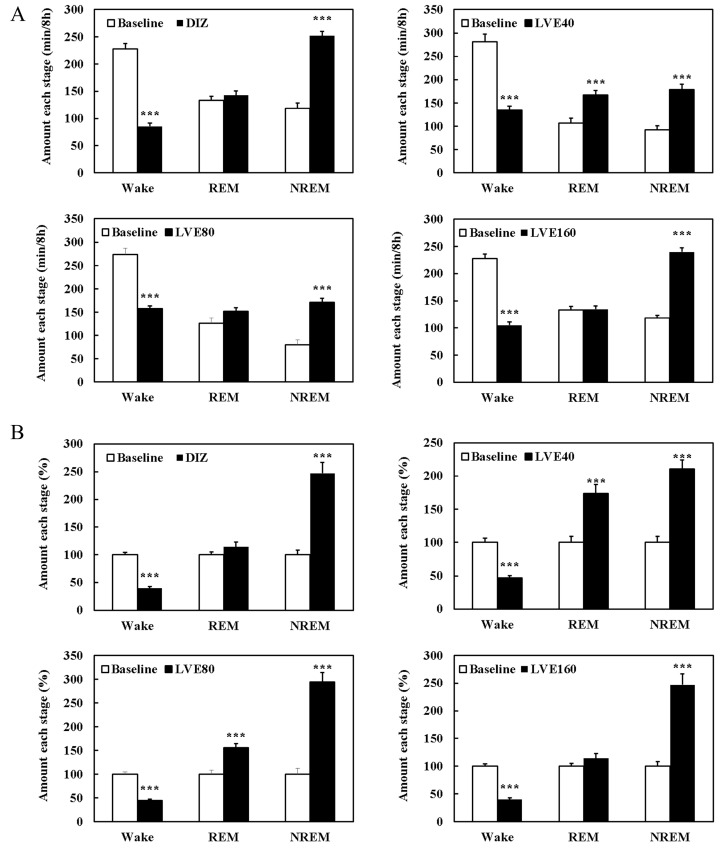
Sleep stage (wake, REM, NREM) duration and its percentage distribution in quantitative EEG analysis. (**A**) Amount each stage (min/8 h). (**B**) Amount each stage (%). Oral administration for each group was continued for 7 consecutive days. EEG recordings were performed for an additional 24 h. Student’s *t*-test was used for comparisons between baseline and each treatment group. Statistical significance is indicated by *** (*p* < 0.001).

**Figure 8 ijms-26-05723-f008:**
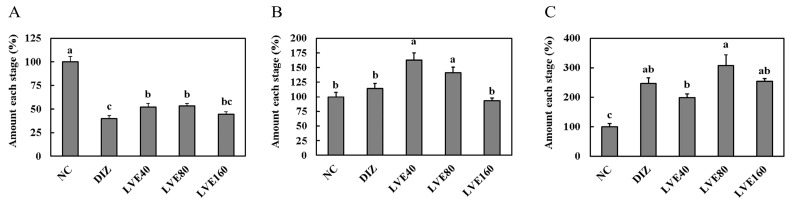
Group-wise comparison of sleep stage percentage distribution (wake, REM, NREM) in quantitative EEG analysis. (**A**) Wake; (**B**) REM; (**C**) NREM. Oral administration for each group was continued for 7 consecutive days. EEG recordings were performed for an additional 24 h. Data are presented as mean ± standard deviation (SD). Statistical significance was determined using one-way ANOVA followed by a post hoc test. Different letters (a, b, c) above the bars indicate significant differences between groups (*p* < 0.05).

**Figure 9 ijms-26-05723-f009:**
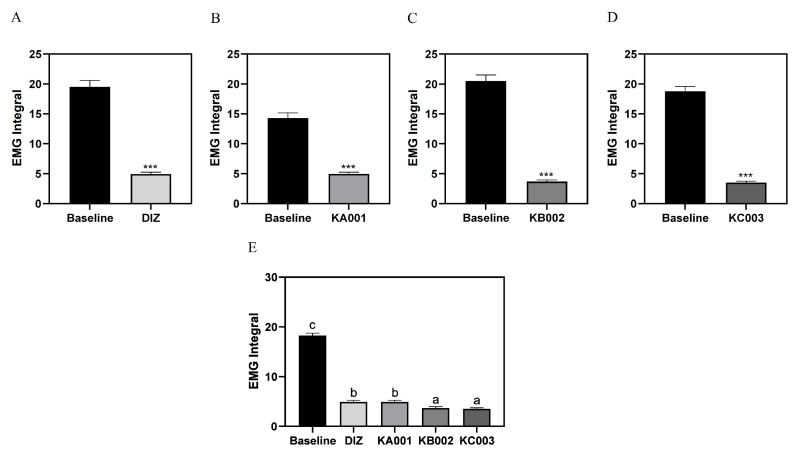
Group-wise comparison of sleep duration on quantitative EMG integral analysis. (**A**) Diazepam 2 mg/kg/day (DIZ); (**B**) LVE 40 mg/kg/day (KA001); (**C**) LVE 80 mg/kg/day (KB002); (**D**) LVE 160 mg/kg/day (KC003); (**E**) group-wise comparison. Oral administration for each group was continued for 7 consecutive days. EMG recordings were performed for an additional 24 h. Data are presented as mean ± standard deviation (SD). Statistical significance was assessed using Student’s *t*-test for comparisons between baseline and treatment samples (*** *p* < 0.001). For comparisons among multiple groups, one-way ANOVA followed by a post hoc test was performed. Different letters (a, b, c) above the bars indicate significant differences between groups (*p* < 0.05).

**Figure 10 ijms-26-05723-f010:**
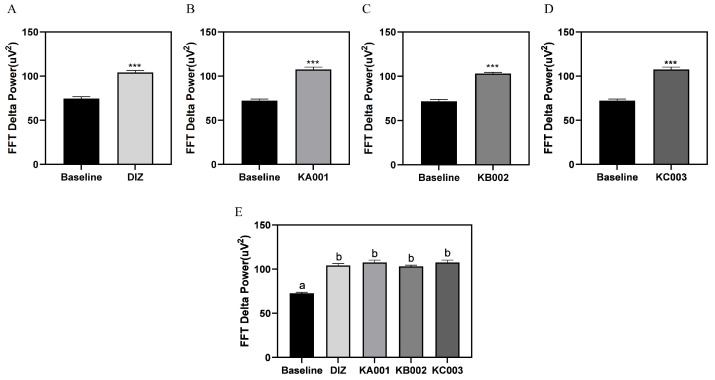
Group-wise comparison of sleep duration on quantitative FFT delta power analysis. (**A**) Diazepam 2 mg/kg/day (DIZ); (**B**) LVE 40 mg/kg/day (KA001); (**C**) LVE 80 mg/kg/day (KB002); (**D**) LVE 160 mg/kg/day (KC003); (**E**) group-wise comparison. Oral administration for each group was continued for 7 consecutive days. Data are presented as mean ± standard deviation (SD). Statistical significance was assessed using Student’s *t*-test for comparisons between baseline and treatment samples (*** *p* < 0.001). For comparisons among multiple groups, one-way ANOVA followed by a post hoc test was performed. Different letters (a, b) above the bars indicate significant differences between groups (*p* < 0.05).

**Table 1 ijms-26-05723-t001:** Body weight changes in C57BL/6J mice related to EEG/EMG-based sleep models.

	DIZ	KA001	KB002	KC003
Body weight (g)	24.0 ± 0.2 ^ns^	24.1 ± 0.2	23.9 ± 0.3	24.1 ± 0.1

DIZ: Diazepam 2 mg/kg, KA001: LVE 40 mg/kg/day, KB002: LVE 80 mg/kg/day, KC003: LVE 160 mg/kg/day. ns: not significant.

## Data Availability

The original contributions presented in this study are included in the article. Further inquiries can be directed to the corresponding author(s).

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
