# Peer review of "Lemon Verbena Extract Enhances Sleep Quality and Duration via Modulation of Adenosine A1 and GABAA Receptors in Pentobarbital-Induced and Polysomnography-Based Sleep Models"

_ijms, 2025, doi:10.3390/ijms26125723_

Round 1

Reviewer 1 Report

Comments and Suggestions for Authors

Reviewer 2 Report

Comments and Suggestions for Authors

How did the authors determine the dose of Lemon verbena extract (LVE)?

How did they optimize the dose of LVE?

What about LD50 of LVE?

What about LD50 of LVE?

What about the pharmacokinetic profiling of LVE?

Why did not they include a separate group for pentobarbital?

Why did they use female mice in some experiment and male mice in another one?

What is the kind of anti-melatonin antibody?

Reviewer 3 Report

Comments and Suggestions for Authors

Aloysia citriodora (A. citriodora) has been traditionally used for its sedative properties and in the treatment of insomnia across various cultures. In the present study, the authors utilized two experimental models to investigate the potential mechanisms through which A. citriodora influences sleep regulation and architecture.

The study is relevant and falls within the scope of the journal. The methodology is clearly described, and the results are discussed in the context of existing literature. The findings contribute to the growing body of evidence suggesting that oral administration of A. citriodora may serve as a complementary treatment for individuals suffering from insomnia. 

Throughout the manuscript, the authors frequently state that they aim to “elucidate” the mechanisms underlying the effects of A. citriodora on sleep. However, it would be more appropriate to state that the study is designed to explore or investigate specific mechanisms, rather than fully elucidate them. Definitive elucidation typically requires more comprehensive mechanistic evidence, including pathway-specific analyses or pharmacological validation. The authors are encouraged to revise such statements throughout the manuscript to adopt a more cautious and scientifically accurate tone when interpreting their findings.

Specific Comments:

Introduction:

  • The introduction does not clearly justify the aim of the study. A stronger rationale should be provided to establish the need for this research.

  • The roles of adenosine, melatonin, and GABA in sleep regulation should be briefly discussed and supported with appropriate references.

  • Some of the content currently presented in the discussion section may be more appropriately placed in the introduction, and vice versa. A reorganization is recommended to improve logical flow.

  • The authors should avoid anticipating the study’s results within the introduction.

  • Claims made in the second and third paragraphs must be substantiated with appropriate citations.

Methods:

  • The authors used ICR mice (female, 4 weeks old) and C57BL/6N mice (male, 9 weeks old). The rationale for using different sexes and strains of mice should be clearly explained.

  • Section 2.2: Clarify the criteria used to determine the doses of LVE and pentobarbital.

  • Section 2.8: EEG recording and sleep-wake state analysis should be supported by relevant references.

  • Section 3.3: The phrase “LVE improves serum melatonin” should be revised. The word “improves” is subjective; “increases” would be more accurate and scientifically appropriate.

Discussion:

  • The discussion is generally well-written. However, given the use of both male and female mice, the authors should address potential sex-related differences in their findings.

  • The integration of the findings with existing literature is adequate, but could be expanded to discuss underlying mechanisms more deeply.

Limitations:

  • The authors should include a section explicitly addressing the limitations of the study, including any factors that may affect the generalizability or interpretation of the results.

Figures:

  • Statistical significance between groups should be clearly indicated under each figure, along with the corresponding p-values or statistical tests used.

.

Comments on the Quality of English Language

  • The manuscript would benefit from a thorough proofreading to correct grammatical, syntactic, and stylistic errors.

Reviewer 4 Report

Comments and Suggestions for Authors

Comments concerning the manuscript ijms-3608359 entitled: “Lemon Verbana extract enhances sleep quality and duration via modulation of Adenosine A1 and GABAA receptors in pentobarbital-induced and polysomnography-based sleep models.” By Mijoo Choi et al.

 General Comments

The aim of the authors was to investigate the effects of Lemon verbana extract (LVE) on sleep regulation using a pentobarbital-induced sleep model and an EEG-based sleep assessment model in mice. The manuscript submitted contains various important drawbacks numbered just below.

Specific Comments

1 - Firstly, we would like to emphasize on the fact that the “pentobarbital-induced sleep model” employed in the study cannot be considered as a “sleep model” but rather as an “anaesthetic animal model.” Pentobarbital is a barbiturate inducing anaesthesia. Genuine biochemical mechanisms of sleep, established throughout several decades of research, have been established different from those of pentobarbital, an anaesthetic opening the chloride channel. Thus, the “pentobarbital-induced sleep model” employed throughout the entire manuscript should be avoided.

2 - Page 2 line 50: After sleep duration, a reference is necessary.

3 - Regarding the pentobarbital-induced sleep in mice (page 2, line 87), the authors employed ICR female mice (four weeks old). However, for the electroencephalographic (EEG) study (page 3, line 133), the authors employed C57BL/6N male mice (nine weeks old). The authors should explain why female and male mice were used in the study. They should also explain why animals at four and nine weeks were employed.

4 – Regarding the sleep latency and duration, it can be considered that this approach was achieved in ICR female mice (to specify). For this study, we strongly believe that the authors were not measuring sleep latency and duration but only the anaesthetic effect of pentobarbital (induction and duration). The authors should accept that barbiturates does not induce physiological sleep.

5 – Regarding the anaesthetic employed, i.e., Avertin (page 3, line 137), why the Ketamine/Xylazine mixture (employed and accepted everywhere) was not currently used?

6 – Regarding the EEG recordings, the information given is not sufficient. Is the “EEG-2 Logger system” (from Japan) an automatic system scoring the sleep states? If this is the case, we are suspicious about the results obtained since an automatic scoring is not accurate enough. Sometimes, rapid eye movement sleep (REM sleep) is not precisely measured. For sleep EEG recordings, a short human visual control is necessary every day. According to the statement of this technological aspect, in the manuscript, we do not know what has been done. Thus, a caution exists for the authenticity of the results obtained. This aspect must be clarified.

7 – Again for the EEG recordings, the authors report (page 4, line 152) that EEG and EMG signals were recorded for 24h starting at 13h00. However, at line 162, they indicate that sleep duration was analysed for an 8-hour period. The protocol employed must be clearer. The situation reported is confusing.

8 – The authors report (page 5, lines 182 to 190) that in pentobarbital-injected mice, LVE induces a notable reduction in sleep latency versus NC group. Moreover, the LVE higher dose (160 mg) also reduces the sleep latency. As suggested before, the results were obtained in animals administered with pentobarbital after LVE injection and only through behavioural observations. The sleep terminology then employed is abusive and the results obtained are dubious.

9 - For the quantitative sleep study achieved, a control group is missing, i.e., animals injected with the solvent only.

10 – Again, for the sleep study a figure with EEG and EMG traces taken during waking, slow wave sleep and REM sleep are necessary.

11 – For the qualitative analysis achieved, the figure 6 is difficult. Results obtained are not conveniently evidenced. Histograms with Delta an EMG power must be shown.

12 - Regarding melatonin (page 12 line 366), the authors consider this transmitter (or neurotransmitter) as a well-known regulator of sleep. This aspect is seriously under caution (see: Psychopharmacology (Berl) 2009, 205: 93-106.  Doi: 10.1007/s00213-009-1519-2. 

Conclusions

The manuscript submitted is issued from an important set of experiments. It can be, however, improved in agreement with the above remarks.

Round 2

Reviewer 1 Report

Comments and Suggestions for Authors

The manuscript is improved. I suggest the author to use * for marking significant differences because their marking is confusing. I agree with the weight monitoring, but I don't see it relevant, because the study didn't aim to assay the influence of the weight on sleep. Also, in the limitation paragraph, I suggest the author to include that they did not compare the LVE with other natural compounds. 

Author Response

Response to Reviewer #1

  1. I suggest the author to use * for marking significant differences because their marking is confusing.

Response: Thank you once again for your valuable feedback.

We understand that using asterisks (*) to indicate statistical significance is a common and intuitive convention, particularly in cases involving comparisons between two groups. However, our study involves multiple-group comparisons, for which ANOVA followed by post hoc testing is the most appropriate and statistically valid method. In such analyses, the use of different alphabetical letters (a, b, c, d) is a standard and widely accepted approach to indicate statistically significant differences among multiple groups.

Additionally, we would like to clarify that our revised manuscript includes expanded data compared to the original submission— In particular, Figures 9 and 10 have been newly added, and these figures include both two-group comparisons and multiple-group comparison graphs. For the two-group comparisons, we used only asterisks (*) to indicate statistical significance. For the other figures involving multiple groups, we have continued to use alphabetical letters to reflect the results of the ANOVA and post hoc tests.

Furthermore, we have taken care to ensure that the use of letters clearly represents the statistical differences. If the letters assigned to groups are not shared (i.e., groups labeled with different letters), this explicitly indicates that the differences are statistically significant. The sequence and variation of these letters are based on the results of multiple comparisons, providing a transparent and rigorous representation of the data.

We believe this method maintains both clarity and statistical integrity, and we hope this explanation addresses your concerns. Please let us know if further clarification is needed.

Thank you again for your thoughtful suggestions and your efforts to improve the quality of our manuscript.

  1. I agree with the weight monitoring, but I don't see it relevant, because the study didn't aim to assay the influence of the weight on sleep.

Response: Thank you very much for your comment regarding the inclusion of body weight monitoring data.

While we understand your perspective that body weight may not be directly related to the primary aim of our study — namely, the investigation of sleep-related outcomes — we respectfully believe that presenting this data is still relevant and valuable. In numerous peer-reviewed studies, including those focusing on disease mechanisms or physiological outcomes not directly associated with body weight, researchers often include basic metabolic indicators such as body weight, glucose, AST/ALT, and creatinine levels. These parameters serve as fundamental physiological markers that help assess the general health status of the animals and the potential systemic effects of the treatment.

In our study, we used a natural compound as the intervention. When working with natural substances, it is particularly important to evaluate whether the compound causes unintended metabolic consequences, such as weight gain or loss. Monitoring body weight allows us to provide initial evidence for the safety of the compound, suggesting that it does not induce obesity or other metabolic disturbances.

Therefore, we believe that the inclusion of body weight data — although not a primary outcome — contributes meaningfully to the overall interpretation of the safety and systemic impact of the treatment. We hope this explanation clarifies our rationale and demonstrates our commitment to presenting a comprehensive and responsible analysis of our experimental results.

  1. Also, in the limitation paragraph, I suggest the author to include that they did not compare the LVE with other natural compounds. 

Response: According to the reviewer’s request, the following sentence was added to the Discussion section (lines 530–536, highlighted as blue color):

"Third, although the current study highlights the beneficial effects of LVE, we did not compare its efficacy with that of other natural compounds known for sleep-promoting properties. Including such comparisons in future studies could offer valuable insights into the relative therapeutic potential of LVE and help position it more accurately within the landscape of natural sleep aids."

Reviewer 2 Report

Comments and Suggestions for Authors

Accept without any further comments.

Author Response

Response to Reviewer #2

Reviewer’s comment: Accept without any further comments.

Response: We sincerely thank the reviewer for their positive evaluation and acceptance of our manuscript. We truly appreciate your time and effort in reviewing our work.

Reviewer 3 Report

Comments and Suggestions for Authors

The authors have taken great care in addressing all my comments thoughtfully and thoroughly.  The revised manuscript has  greatly improved; it is much clearer, better organized, and makes a stronger contribution to the literature. I have no further substantive concerns.

Author Response

Response to Reviewer #3

Reviewer’s comment: The authors have taken great care in addressing all my comments thoughtfully and thoroughly.  The revised manuscript has greatly improved; it is much clearer, better organized, and makes a stronger contribution to the literature. I have no further substantive concerns

Response: We sincerely thank the reviewer for their positive evaluation and acceptance of our manuscript. We truly appreciate your time and effort in reviewing our work.

Reviewer 4 Report

Comments and Suggestions for Authors

Quite all my suggestions have not been taken into account.

Now, the editor should decide what to do whith your manuscript.

Author Response

Response to Reviewer #4

We sincerely thank Reviewer 4 for the thoughtful and detailed comments provided. While the reviewer has expressed concern that the suggestions may not have been fully reflected in the manuscript, we would like to take this opportunity to express our special gratitude to Reviewer 4 in particular. Among all the valuable feedback we received, it was Reviewer 4’s insightful suggestions that led to the most substantial revisions in our manuscript. In response to the reviewer’s comments, we incorporated significant modifications, including the addition of new data and figures, which have greatly improved the clarity and scientific rigor of our work.

* The following are the revisions we made in response to the reviewer’s comments.

#2. Missing references — We have added the missing references accordingly (page 2, line 58).

#3. Regarding the reviewer’s comments on the animal model — We provided detailed explanations about the sex and age of the mice used in the experiments, which are clearly described in the Methods section (2.2 and 2.7).

#5. Regarding the use of Avertin — Based on previous literature, Avertin (tribromoethanol) was used during EEG electrode implantation surgery due to its rapid recovery and minimal physiological side effects. This choice enables stable anesthesia maintenance and quick recovery, minimizing its impact on sleep measurements. Ketamine/xylazine was avoided because it has been reported to cause longer recovery times and potentially affect EEG waveforms and biological rhythms.

#6. Regarding the reliability of EEG sleep scoring and whether only automated classification was used — We provided an appropriate explanation, which is reflected in the Methods section (2.8).

#7. Regarding confusion about the EEG analysis period — EEG/EMG recordings were conducted continuously for 24 hours, with focused analysis on the physiologically meaningful 8-hour dark phase (active period), which is a standard approach in many sleep studies. The data were analyzed in 10-second epochs and presented in 1-hour intervals. These details have been clarified in the Methods section (2.8).

#9. Regarding the absence of a vehicle-only control group in EEG sleep analysis — We responded that the EEG/EMG sleep experiments were designed using a within-subject approach, where each individual served as its own control.

#10. Regarding the reviewer’s request for examples of EEG waveforms — We have added representative EEG waveform examples to Figure 6 and revised the figure accordingly.

#11. Regarding the reviewer’s comment on qualitative analysis, specifically that Figure 6 was difficult to interpret and that histograms of delta and EMG power should be provided — We have included specific graphs of integrated EMG values and FFT-based delta power data as new Figures 9 and 10, respectively.

#12. Regarding the reviewer’s caution on the explanation of melatonin — We revised the interpretation of melatonin’s role more cautiously, citing relevant recent literature to reflect the latest research findings and ongoing discussions within the scientific community. (These revisions are reflected on page 15, line 465.)

**The following points are those for which we did not incorporate the reviewer’s suggestions.

Regarding the reviewer’s comments on the pentobarbital-induced sleep model (#1, #4, #8), the reviewer repeatedly suggested that the use of the term “sleep” may be inappropriate. However, we respectfully did not follow this suggestion because we believe that the term “sleep” is necessary for this study.

We acknowledge that the use of the term “sleep” reflects an anesthetized state and recognize its limitations. Accordingly, we have clearly stated in both the Introduction and Discussion sections that the pentobarbital model serves as a convenient tool for screening vestibular and sleep-inducing effects, but it does not fully replicate the complex mechanisms of natural sleep.

Furthermore, to strengthen the validity and physiological relevance of our findings, we conducted EEG/EMG analyses to measure sleep onset time, total sleep duration, as well as REM and NREM sleep architecture.

Finally, to maintain consistency with existing literature and adequately explain the behavioral and electrophysiological indicators observed in this study, we have retained the use of the term “sleep.”

We would also like to emphasize that the results of this study using the LVE natural product may provide important supporting evidence for the future development and regulatory approval of health functional foods aimed at improving sleep.